# ADDRESSING SOME LIMITATIONS OF TRANSFORMERS WITH FEEDBACK MEMORY

## ABSTRACT

Transformers have been successfully applied to sequential, auto-regressive tasks despite being feedforward networks. Unlike recurrent neural networks, Transformers use attention to capture temporal relations while processing input tokens in parallel. While this parallelization makes them computationally efficient, it restricts the model from fully exploiting the sequential nature of the input. The representation at a given layer can only access representations from lower layers, rather than the higher level representations already available. In this work, we propose the Feedback Transformer architecture that exposes all previous representations to all future representations, meaning the lowest representation of the current timestep is formed from the highest-level abstract representation of the past. We demonstrate on a variety of benchmarks in language modeling, machine translation, and reinforcement learning that the increased representation capacity can create small, shallow models with much stronger performance than comparable Transformers.

## 1 INTRODUCTION

In recent years, the Transformer architecture (Vaswani et al., 2017) has brought large improvements to a wide range of Natural Language Processing tasks such as machine translation, sentence representation (Devlin et al., 2019), and summarization (Edunov et al., 2019). Transformers are also successfully used as an autoregressive model on sequential tasks such as language modeling (Dai et al., 2019; Rae et al., 2020) and reinforcement learning (Parisotto et al., 2019). Unlike more traditional recurrent architectures such as RNNs and LSTMs, the Transformer architecture processes a sequence in parallel in an order-invariant way. Techniques such as position embeddings (Sukhbaatar et al., 2015; Shaw et al., 2018) and attention masking are required to capture input order information. In this work, we focus on several limitations of the Transformer architecture as an autoregressive model and present a straightforward solution — *Feedback memory*. These limitations and our proposed solution target sequential token prediction tasks, such as language modeling or other auto-regressive generative tasks.

The feedforward nature of Transformers makes them efficient on modern hardware, but restricts the Transformer from taking full advantage of the input's sequential property. In particular, the current hidden representation of a Transformer only accesses the past representations of lower layers, even though higher level representations of the past have already been computed as an autoregressive model. At generation, the Transformer generates only one token at a time, so it could access these representations for better performance, but does not exploit these at training time due to parallelization. However, if these past higher level representations could be used at training time, they would enrich future lower level representations, enabling shallower models to have the same representation power.

Another inherent limitation of Transformers on sequential tasks is the lack of recursive computation (Dehghani et al., 2018), and the number of transformations possible on the input is bounded by the model depth. Such disadvantages have impact on tasks that require careful tracking of a world state or modeling hierarchical structures (Tran et al., 2018; Hahn, 2020). On the other hand, while RNNs can maintain an internal state for an unbounded time while accumulating more computations upon it, the size of this internal state is limited by the dimension of the hidden state.

In this work, we propose a novel autoregressive model, the *Feedback Transformer*, that makes all previous hidden representations accessible to the computation of a representation at any depth — the model *feeds back* previous computations to itself. The feedback allows the model to perform

recursive computation, building stronger representations iteratively upon previous states. To achieve this, we modify self-attention to attend to higher level representations rather than lower ones.

As shown in Figure 1, the Feedback Transformer merges the hidden states from all layers into a single vector for every time step and stores them in a memory. Instead of self-attention, all subsequent layers attend to this memory, which means every previously computed representation is accessible by all future layers, mediated by the memory. This allows Feedback Transformers to recursively compute and transform an input as many times as the input length, which is something Transformers cannot achieve. While RNNs can perform recursive computation, the amount of information that Feedback Transformers can maintain is not limited by the number of layers.

There are computational benefits to this straightforward modification. First, it uses less memory because all the layers share a single Feedback memory, thus reducing the memory size by $L$ times, where $L$ is the number of layers. There is also less computation because we share the key and value projections during attention computation, which increases the speed of the attention over the Feedback Memory. Further, the GPU memory usage is reduced due to the memory sharing — the overall model is 2x smaller — allowing the batch size to be increased for computational efficiency. During inference, the increased batch size contributes to substantially faster decoding speeds.

In summary, our main contributions are: (1) The Feedback Transformer architecture, which completely changes the way a Transformer works to access available higher level representations immediately. (2) We show the Feedback Transformer can achieve state of the art results with smaller, shallower models that have faster decoding speed and smaller memory footprint. (3) The Feedback Transformer uses substantially less memory during training and inference time.

## 2 RELATED WORK

Several previous works have analyzed the limitations of Transformer architectures, such as the inability to process input sequentially (Dehghani et al., 2018) or represent hierarchical structure (Tran et al., 2018). Hahn (2020) demonstrate that Transformers cannot model structures involving bounded recursion, such as closing parentheses. Pérez et al. (2019) study Transformers in the context of Turing machines, where they must produce unbounded numbers of decoding steps. Various work in probing Transformers identified several limitations where Transformers may not have the computational capacity of recurrent architecture like an LSTM (Hahn, 2020).

From the architectural perspective, our work shares similarities with recurrent networks augmented with external shared memories (Graves et al., 2014; Joulin & Mikolov, 2015; Sukhbaatar et al., 2015). For example, the stack augmented RNN of Joulin & Mikolov (2015) adds an external memory to a recurrent network to keep long term dependencies. Closer to our work, the Neural Turing Machine of Graves et al. (2014) models an unconstrained memory that resembles the self-attention layer of a Transformer. Further improvements to recurrent networks, such as the Gated Feedback RNN (Chung et al., 2015), are based on better controlling signal from different layers and extended to feedback through multiple pathways (Jin et al., 2017). These works are built on recurrent networks with additional components to store long term dependencies.

Other works have studied modifications to the Transformer architecture by enriching its structure with components inspired by recurrent networks. For example, Wang et al. (2019) propose adding a local recurrent sublayer to the Transformer layer to remove the need for position embeddings in the multi-head self-attention layers. Universal Transformer (Dehghani et al., 2018) share the parameters between the layers of a Transformer, leading a recurrent network in depth. Hao et al. (2019) and Chen et al. (2018) augment Transformers with a second, recurrent encoder. As opposed to our work, these prior investigations do not change the computational path in a Transformer to reduce the discrepancy between the training and inference time. Closer to our work, Merity (2019) proposes adding a self-attention layer on top of the past outputs from an LSTM cell. However, this approach keeps the recurrent and the self-attention mechanisms decoupled, as opposed to ours which makes the attention mechanism recurrent. In particular, the LSTM layer of Merity (2019) still intrinsically has a bottleneck corresponding to the dimension of the hidden layer.

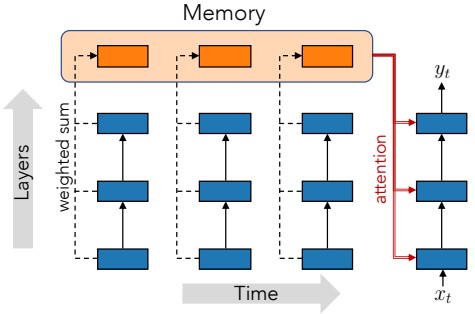

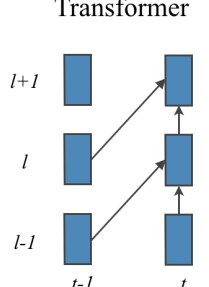

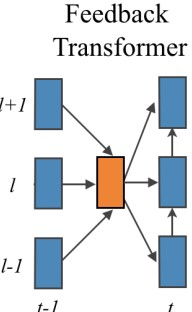

Figure 1: The **Feedback Transformer** merges past hidden representations from all layers into a single vector and stores it in memory.

Figure 2: **Difference between Feedback and Transformer.** $t$ indicates the timestep and $l$ indicates the layer.

## 3 METHOD

In this section, we propose the Feedback Transformer, which provides capacity to build richer representations of each timestep $t$ of a sequential modeling task.

### 3.1 TRANSFORMER ARCHITECTURES

We briefly describe the Transformer (Vaswani et al., 2017). Each layer is composed of a multi-head self-attention sublayer (`Attn`) followed by a feedforward sublayer (`FF`), and each sublayer is followed by an add-norm operation that combines a skip-connection (He et al., 2016) and layer normalization (Lei Ba et al., 2016). The $l$-th layer of a Transformer processes an input sequence of vectors $\mathbf{X}^l = (\mathbf{x}_1^l, \ldots, \mathbf{x}_t^l)$ into a sequence of vectors of the same length. First, the self-attention sublayer computes a representation for each time step $t$ by taking its related input vector $\mathbf{x}_t$ along with its past context, $\{\mathbf{x}_{t-\tau}^l, ..., \mathbf{x}_{t-1}^l\}$:

$$\mathbf{z}_t^l = \mathtt{Attn}(\mathbf{x}_t^l, \{\mathbf{x}_{t-\tau}^l, \ldots, \mathbf{x}_{t-1}^l\}).$$

Within the self-attention sublayer, $\mathbf{x}_t^l$ is used to form query vectors while its context is used to compute key and value vectors, forming a memory of the past information. Then the feedforward sublayer processes each vector $\mathbf{z}_t^l$ independently, i.e., $\mathbf{x}_t^{l+1} = \mathtt{FF}(\mathbf{z}_t^l)$. The Transformer layer transforms its input sequence into an output sequence $\mathbf{X}^{l+1} = \mathtt{FF}(\mathtt{Attn}(\mathbf{X}^l))$.

In practice, a block of steps $\{x_{t-M+1}^l, \ldots, x_t^l\}$ is computed in parallel during training, where $M$ can be seen as the backpropagation through time (BPTT) length. This makes training Transformers efficient on hardware such as GPUs. However, to operate on sequences of unbounded length, Transformers require modifications such as caching and relative position embeddings (Dai et al., 2019; Sukhbaatar et al., 2019).

### 3.2 LIMITATIONS OF TRANSFORMERS

Previous work has analyzed the impact of several limitations of the Transformer architecture, such as the inability to track long sequences and process hierarchical inputs (Hahn, 2020). In this work, we focus on two major limitations of Transformer architectures.

**Limited Access to Higher Level Representations.** Layer by layer, Transformers build more abstract, high level representations of the input sequence. At each layer, the representations for the input sequence are treated in parallel. As a consequence, a Transformer does not leverage the highest level representations from the past to compute the current representation, even though these highest level representations have *already been computed* for autoregressive models.

**Maintaining a Belief State.** Many sequential tasks require models to maintain an internal state for two main purposes. First, internal states act as memory for recalling past inputs, where Transformers excel because their internal state $x_t^l$ is directly accessible to future steps through self-attention.

The second role of an internal state is to act as a belief state that tracks the world state that is not directly observable in inputs. For example, when inputs are actions taken on a Markov Decision Process, an internal state can apply those changes to the current belief state and correctly predict the outcome. As a feedforward model, Transformer have inherent limitations in this area — only a fixed number of transformations can be applied to its internal states. Since both `Attn` and `FF` sublayers contain a fixed number of transformations and there are $L$ layers of them, the total number of transformations between the input and output is limited by the depth. This means Transformers cannot maintain an internal state for long time if it has to be frequently updated.

### 3.3 FEEDBACK TRANSFORMER

We propose to change the Transformer architecture by using the most abstract representations from the past directly as inputs for the current timestep. This means that the model does not form its representation in parallel, but sequentially token by token. More precisely, we replace the context inputs to attention modules with memory vectors that are computed over the past, i.e.,

$$\mathbf{z}_t^l = \texttt{Attn}(\mathbf{x}_t^l, \{\mathbf{m}_{t-\tau}, \ldots, \mathbf{m}_{t-1}\}),$$

where memory vectors $\mathbf{m}_t$ are computed by summing the representations of all layers at time step $t$:

$$\mathbf{m}_t = \sum_{l=0}^{L} \texttt{Softmax}(w^l)\mathbf{x}_t^l, \tag{1}$$

where $w^l$ are learnable scalar parameters. Note these scalars are the only new parameters introduced by our change, with all else the same as the standard Transformer. Here $l = 0$ corresponds to token embeddings. The weighting of different layers by a softmax output gives the model more flexibility as it can average them or select one of them.

This modification of the self-attention input adapts the computation of the Transformer from parallel to sequential, summarized in Figure 2. Indeed, it provides the ability to formulate the representation $\mathbf{x}_{t+1}^l$ based on past representations from any layer $l'$, while in a standard Transformer this is only true for $l' < l$. This change can be viewed as exposing all previous computations to all future computations, providing better representations of the input. Such capacity would allow much shallower models to capture the same level of abstraction as a deeper architecture. This has several practical advantages, as more shallow models have reduced memory footprint and increased decoding speed.

An alternative view of such an architecture modification is providing the capacity for recursive computation — outputs from a sublayer can feed back to the same sublayer through the memory. The model can then maintain an internal state for unbounded time. This is a clear advantage over Transformers, in which a submodule never looks at its own output. While an RNN can also repeat its computation on its internal state, its internal state has a limited capacity determined by the number of layers and their hidden dimension. In contrast, the internal state of a Feedback Transformer is its whole memory, which can grow with the input length. This allows the model to keep track of a large number of things within its internal state.

While our modification requires sequential computation, we significantly improve training speed by sharing the key and value projections $W_k^l$ and $W_v^l$ across all layers. This sharing reduces computation because we need to compute key and value vectors only once instead of computing them per layer

$$\mathbf{k}_t^l = \mathbf{k}_t = W_k\mathbf{m}_t \quad \mathbf{v}_t^l = \mathbf{v}_t = W_v\mathbf{m}_t.$$

For the same reason, the memory footprint is smaller than a standard Transformer because only one set of $\mathbf{k}_t, \mathbf{v}_t$ needs to be stored. To be more precise, the memory requirement for processing a single token is reduced from $O(L \times T)$ to $O(T)$, where $L$ is the number of layers and $T$ is the context size. Further, the reduced memory usage allows the batch size to be increased to recover some of the lost parallelism, which improves training speed. Thus, the Feedback Transformer is not much slower compared to the standard Transformer. Note that the same sharing of projections will not make the standard Transformer efficient because those projections are applied to different representations at each layer (the key and value vectors will not the same for all layers).

Lastly, we note that the sequential nature of the Feedback Transformer does not affect the performance during generation where one needs to compute one step at a time anyway. The same is true for online reinforcement learning where the input must be processed sequentially even during training.

| Task / Model | Accuracy (%) | |
|---|---|---|
| **Copy** | **Char** | **Seq** |
| Transformer | 59.1 | 6.2 |
| Feedback Transformer | 76.2 | 23.6 |
| **Reverse** | **Char** | **Seq** |
| Transformer | 50.2 | 5.9 |
| Feedback Transformer | 74.8 | 29.2 |
| **Counting** | **Len 50** | **Len 1K** |
| Transformer | 99.6 | 82.4 |
| Feedback Transformer | 99.7 | 95.3 |
| **Random Walk** | | |
| Transformer | 68 | |
| Feedback Transformer | 100 | |
| **Algorithmic Task** | **3 vars** | **5 vars** |
| Transformer 4L | 33.7 | 37.5 |
| Transformer 8L | 47.4 | 29.1 |
| LSTM | 82.8 | 32.1 |
| Feedback Trans. 4L | 99.1 | 92.6 |

Table 1: **Results on toy tasks.** Char is character accuracy, Seq is sequence accuracy.

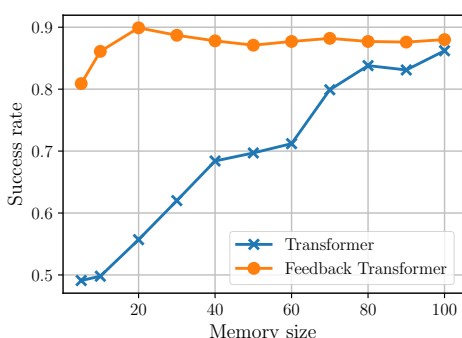

Figure 3: **Results on the Corridor task.** The Transformer degrades as the memory size decreases, but the Feedback Transformer maintains performance.

## 4 EXPERIMENTS

We explore different sequential input tasks in natural language processing and reinforcement learning. First, we demonstrate the downsides of the standard Transformer architecture on tasks where the Transformer performs poorly. We show that the Feedback Transformer is able to overcome challenges and retain long memory. Next, we highlight the strength of the Feedback architecture in building complex, high level representations even with shallow models. We demonstrate that the Feedback model can achieve significantly stronger results than Transformer models, an effect that is exaggerated as models get smaller. Finally, we compare the Feedback architecture to the Transformer architecture with other work on standard long-context language modeling tasks. In experiments on large datasets, we use the shared key-value projections to improve training time. Additional experimental details and results can be found in the appendix.

### 4.1 LIMITATIONS OF TRANSFORMER: ILLUSTRATIVE TASKS

#### 4.1.1 LIMITED ACCESS TO LONG MEMORY

First, we examine the Transformer's limited access to long memory on several simple, straightforward tasks that illustrate this. Unlike the standard Transformer, the Feedback architecture is able to remember information over many timesteps.

**Walking down a Corridor.** In this reinforcement learning task, each agent is placed at the start of a long corridor with either a *blue* or *green* object. The agent must look at the object's color, walk down the corridor, and go through the corresponding colored door at the end. The only task is to remember the color and not become distracted by walking down the very long hallway. Results are shown in Figure 3 and show that the performance of the Transformer degrades quickly as the memory size shrinks, but the Feedback Transformer maintains strong performance at all memory sizes.

**Copy and Reverse.** We experiment next on two algorithmic tasks, copy and reverse (Kaiser & Sutskever, 2015). We train on sequences of length 40 consisting of integers 0 through 9, and test on sequences of length 400. Models read the input and then either copy or reverse, which requires memory over the sequence and the ability to track position, as well as generalization capability as the train and test settings are different lengths. We consider two variations of copying and reversing:

either at the character level or at the sequence level. Results are shown in Table 1. The Feedback architecture has large improvements in accuracy, indicating improved memory and positional tracking.

**Counting.** Finally, we experiment on a counting task, where models have a sequence of *A*'s in a row, and must output the corresponding quantity of the letter *B*. The model must count the number of the A's to output the correct number of B's. We consider two settings: training on short sequences of lengths up to 50 and training on long sequences of lengths up to 1000. We show results in Table 1, where we demonstrate the Feedback model is much better at counting over long sequences.

### 4.1.2 LIMITED STATE UPDATES

The complexity of the representations the Transformer is able to formulate is strictly dependent on the depth, as each layer of the Transformer allows for additional nonlinearity. The Transformer, then, can only update its state the same number of times as it has layers. We demonstrate that the Feedback Transformer does not have this limitation — in tasks where the model must carefully track and update its state, the Feedback architecture is able to update its state at each timestep.

**Random Walk.** We consider a random walk in a small grid where actions are: go forward 1 step, left turn, and right turn. Given a history of actions and the agent's initial position, it is strictly possible to calculate the current position. The task is trivial because a human could write down the current location and direction and keep updating with each action. However, Transformers cannot do this because they lack a storage that can be updated with each input. Its hidden state can store this information, but with each update, that information has to go up one layer.

An alternative approach to this task is to solve it all at once given a sequence of actions, which is feasible for Transformers since they can access all inputs with their attention. However, this approach is challenging because the effect of each action depends on the direction at that point and whether the agent is on the edges, which itself is not known yet. This can be seen in Table 1, where the Transformer struggles and only reaches $68\%$ accuracy. In contrast, the Feedback Transformer achieves $100\%$ accuracy, which indicates the ability to track state for a long period of time. Both models are trained on 10K sequences, each containing 100 random actions and positions.

**Algorithmic task.** A more complex setting where tracking and updating of a state is crucial is code executions. A model needs keep track of all variable values and update them if necessary. To demonstrate this, we create a simple algorithmic task that consists of the following simple statements: assignments (e.g. `x=5`), increments and decrements (e.g. `y--`), conditionals (e.g. `if x==4: y++`), and print commands (e.g. `print(x)`). Each task consists of 100 randomly selected statements. We consider two settings with 3 and 5 different variables.

Processing of each statement in parallel will not work because conditional statements cannot be executed without knowing the current variable value, which itself can depend on another conditional. As shown Table 1, Transformers cannot solve this task because every time a variable increment or decrement, its value can only be found one layer up in the model, and eventually will be lost. Doubling their layers does help little, but their accuracy is far from perfect. A recurrent model like LSTM is capable of storing a variable value while updating it, thus perform well on the 3 variables version. However, its performance drop when there are more variables because it has to store all their values in a single vector. The Feedback Transformer does not have this bottleneck, and can access updated variable values from the lowest layer, so it gives strong performance on this task.

### 4.2 ADVANTAGES OF FEEDBACK ARCHITECTURE

We examined two limitations of standard Transformers that we improve upon: limited memory span and limited ability to update state. In the Feedback model, we improve on these limitations and now analyze performance on practical tasks including translation and reinforcement learning.

### 4.2.1 STRONG PERFORMANCE WITH SMALL, SHALLOW MODELS

The Feedback Transformer is able to create higher level, more abstract representations with fewer layers and less capacity, as a layer can use all of the most recently created representations of previous

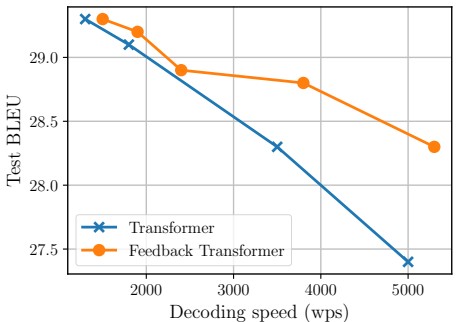 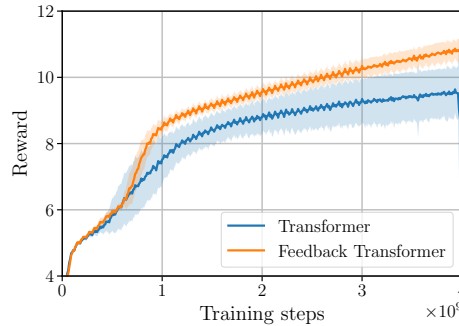

Figure 4: **(left) Machine Translation on `WMT14 En-De`,** test set BLEU and decoding speed in words-per-second for varying decoder depths. **(right) Maze Navigation in Gridworld.** We display average reward comparing Feedback Transformer to standard Transformers.

timesteps. We demonstrate on neural machine translation that the Feedback model performs much better than Transformers at small, shallow sizes. Note that for sequence to sequence, we use Feedback Transformers only in the decoder because the encoder inputs are available simultaneously.

We evaluate the performance of the Feedback Transformer on the `WMT14 En-De` machine translation benchmark of 4.5 million pairs. We follow Vaswani et al. (2017) and train on `WMT16` using `newstest2013` as dev and `newstest2014` as test. We learn 32K joint byte pair encodings (Sennrich et al., 2016), generate with beam size 5, tuning a length penalty on the dev set. We average the last 10 checkpoints and apply compound splitting and compute tokenized BLEU.

In Figure 4 (left), we display results when making the model *shallower only* — layers are removed from a Feedback Transformer decoder compared to Transformers. As the decoder becomes shallow, the gap in performance between the two architectures widens. While the 1-layer Transformer model can only reach 27.3, the Feedback Transformer has 28.3 BLEU. Shallow decoders are critical to fast inference — reducing to 1-layer improves decoding speed by 4.2x, while only losing 1 BLEU with the Feedback architecture. Such results are useful for practical applications, where the speed of producing a translation is very important. We report decoding speed in tokens per second on 1 GPU.

We further experiment with large encoder but shallow decoders. The Feedback Transformer achieves **29.0** BLEU with 12 layer encoder and 2 layer decoder. As the encoder is parallelized even during inference, the increased size of the encoder has negligible impact on decoding speed. To stabilize the training of deeper models, we use LayerDrop (Fan et al., 2019).

### 4.2.2 LONG MEMORY TRACKS STATE

We apply Feedback to a reinforcement learning maze task that requires long memory to optimally solve because agents have limited vision. Note that in such reinforcement learning tasks, the models are trained online using A2C, so the input must be processed sequentially even during training time. Thus, the non-parallelized nature of the Feedback Transformer is not a drawback, and training Feedback Transformers is as fast as Transformers.

The goal is to navigate a procedurally generated random maze where colored objects are placed. One of the colors will be randomly selected as a target, and the agent has to reach it for a reward and a new target. For optimal performance, the agent must remember the maze and object locations. In addition, the agent has turn actions like the Random Walk task, which makes it necessary to keep track of its location and orientation. As shown in Figure 4 (right), the Feedback Transformer converges to reach higher average reward, compared to Transformers. Results are shown averaged over 10 trials.

### 4.3 COMPARISON TO OTHER ARCHITECTURES

In this section, we first, we compare Feedback to recurrent architectures such as LSTM, as well as hybrid RNN-Transformer architectures, and show that the Feedback is more powerful than recurrence

| Model | Test |
|---|---|
| **Recurrent Architectures** | |
| DenseNMT Shen et al. (2018) | 25.5 |
| RNMT+ (Chen et al., 2018) | 28.5 |
| **Hybrid Architectures** | |
| BiARN (Hao et al., 2019) | 28.9 |
| SRU (Lei et al., 2017) | 28.4 |
| **Transformer Architectures** | |
| Transformer (Vaswani et al., 2017) | 28.4 |
| Transformer (Ott et al., 2018) | 29.3 |
| Feedback Transformer | 29.5 |

Table 2: **Results on** `WMT En-De` comparing the Feedback Transformer to Recurrent architectures, hybrid Recurrent-Transformer models, and standard Transformers.

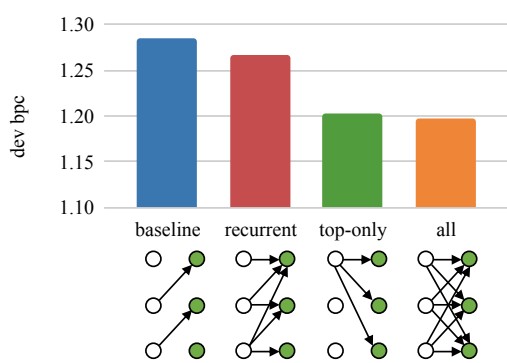

Figure 5: Comparison of different memory composition strategies on `char-PTB`. The recurrent connection alone is not as effective as feedback connections from a higher layer.

alone. Next, we compare our construction of the Feedback Memory with other possible compositions. Lastly, we compare to other Transformer architectures on competitive benchmarks.

### 4.3.1 COMPARISON TO RECURRENT ARCHITECTURES

We compare the Feedback Transformer architecture to recurrent architectures like LSTMs as well as hybrid RNN-Transformer architectures. In Table 2, we display that the Feedback Transformer has stronger performance than the Transformer, RNN, and RNN-Transformer hybrid model. We note that recurrent models address some limitations of Transformer architectures, but the Feedback mechanism goes beyond that. By allowing all past representations to be immediately available for the computation of future representations, Feedback is stronger than Recurrence alone — Recurrent models can only see representations from the previous layer (as depicted in Table 2).

### 4.3.2 MEMORY COMPOSITION

We next investigate the importance of the specific memory mechanism of the Feedback architecture on `char-PTB`. The Feedback architecture uses all layers when creating the memory, motivated by providing access to the entire past of all computations, but other ways of creating the memory as possible. For example, Recurrent architectures have a different memory structure. In multi-layer RNNs, each layer has recurrent connections to the same layer, but not to higher layers. This is an advantage of Feedback architectures — even the highest level abstractions are immediately available.

In Figure 5, we examine the construction of the Feedback memory, comparing our choice of making *all* computation accessible with *recurrent* memory that can access all previous layers plus the same layer, and *top-only* memory that can attend only to the topmost layer. The Feedback Transformer has the best performance, closely matched by *top-only* memory. This indicates the importance of high level representations (see Appendix 6.4 for further analysis on this). Note that recurrence alone is not enough for good performance, and thus the Feedback memory provides richer representations beyond the capacity of recurrent networks.

### 4.3.3 COMPARISON TO OTHER TRANSFORMER ARCHITECTURES

Finally, we examine the performance of Feedback Transformer on long context language modeling benchmarks. We use caching (Dai et al., 2019) and relative position embeddings. Mechanisms applied at inference time (Khandelwal et al., 2019; Krause et al., 2019) can further improve all models, so we do not focus on these.

**Wikitext-103.** We evaluate on word-level language modeling on `Wikitext-103` (Merity et al., 2017). Our Feedback architecture takes 3.5 days to train, compared to the Transformer which takes 1.2 days. We train a small Feedback model, about half the size of Transformer-XL, and find

| Model | Params | Test |
|---|---|---|
| Best Existing (Roy et al., 2020) | — | 15.8 |
| Trans-XL (Dai et al., 2019) | 257M | 18.3 |
| Our Transformer | 140M | 19.9 |
| Feedback Transformer | 126M | 18.3 |

Table 3: **Results on `WikiText-103`.** We report perplexity on test.

| Model | Params | Test |
|---|---|---|
| Best Existing (Rae et al., 2020) | 277M | 0.97 |
| Trans-XL (Dai et al., 2019) | 277M | 0.99 |
| Feedback Transformer | 77M | 0.96 |

Table 4: **Results on `Enwiki8`.** We report bit-per-byte on test.

that it can match the performance of Transformer-XL, as shown in Table 3. This indicates the additional representational capacity of Feedback memory. If we train a standard Transformer that is approximately the same size as our Feedback Transformer, we find it has worse performance (19.9 PPL rather than 18.3). Further, mechanisms like the Routing Transformer can be added to the Feedback Transformer as well. We focus on starting with Transformer-XL as a baseline and showing we can match the performance with a much smaller model.

**Enwiki8.** Finally, we test our model on character-level language modeling in `Enwiki8` (Mahoney, 2011), containing 100M unprocessed bytes from Wikipedia. We train a relatively small 12-layer model, that is one third of the size of the Transformer-XL baseline. Since the task requires very long context, we use adaptive attention span (Sukhbaatar et al., 2019). As shown in Table 4, the Feedback Transformer model achieves a new SOTA performance of 0.96 bit-per-byte despite its small size.

## 5 CONCLUSION

We propose a novel reformulation of the Transformer that fully exploits sequential input — the increased representation power and recursive computation of the Feedback Transformer allows shallow and small models to have much stronger performance compared to a Transformer of the same size. This architecture addresses two fundamental limitations of Transformers as an autoregressive model — limited access to long memory and limited ability to update state. We demonstrate on a variety of tasks the advantages of the Feedback architecture to illustrate the strong performance of this straightforward modification.

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

## 6 ADDITIONAL RESULTS

### 6.1 REINFORCEMENT LEARNING

**Maze Navigation Easy.**   We experiment with a slightly different version of the Maze Navigation task. Instead of an agent with forward, turn-left and turn-right actions, the agent has no orientation and there are only 4 movement actions corresponding to 4 cardinal directions. This makes navigation easier because the agent do not need to keep track of its orientation. Further, it is much easier to compute relative locations given a history of actions. This might explain why standard Transformers are not far behind Feedback Transformers in performance as shown in Figure 6 (left). We also compare to LSTMs, which performs much worse. See Section 7.2 for more implementation details.

**Water Maze.**   We modify the Morris Water Maze task (Morris, 1981) to make it more challenging. The maze is defined by a goal position and a mapping of cell to ID — these remain fixed within an episode but change between episodes. The agent receives as an observation the cell IDs of its current location and the target cell. When the agent finds the target, it receives +1 reward and is randomly teleported. During the same episode, if the agent reaches a previously seen cell, it needs to remember how it reached the target from there to go back. Results are shown averaged over 10 trials (the reward is reported averaged over the last 500 episodes for each trial). As shown in Figure 6 (right), the Feedback Transformer converges to higher average reward.

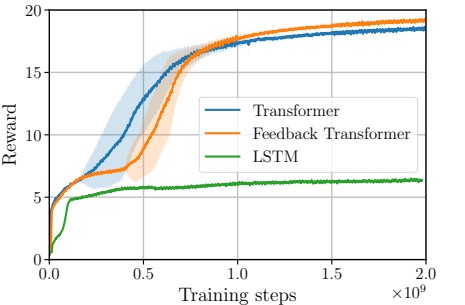 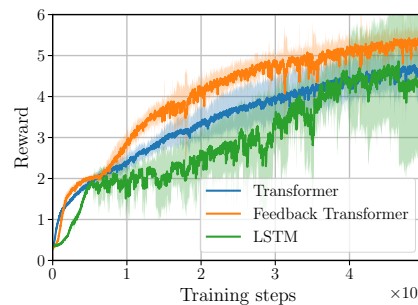

Figure 6: Averaged cumulative reward during training on **(left) Maze Navigation Easy** and **(right) Water Maze** tasks.

### 6.2 IWSLT DE-EN

We additionally evaluate the Feedback Transformer on `IWSLT De-En`, a small machine translation dataset. We train a small Transformer model with 6 layers. For generation, we use beam size 5 without checkpoint averaging. Model quality is evaluated using tokenized BLEU. Results are shown in Figure 7 (left) and show that for shallower models, the Feedback Transformer has better performance than the standard Transformer.

### 6.3 SUMMARIZATION ON CNN-DAILYMAIL

We evaluate on the CNN-Dailymail multi-sentence summarization benchmark of 280K news articles Hermann et al. (2015), modeling the first 400 words of the article See et al. (2017). We evaluate using ROUGE Lin (2004). and use 3-gram blocking and tune length Fan et al. (2017). Figure 7 (right) displays the performance of the Feedback Transformer as the decoder layers are reduced, making the model *shallower only*. For all model depths, the Feedback architecture maintains a consistent improvement in ROUGE compared to the standard Transformer. Compared to sentence-level tasks such as translation, this summarization benchmark requires multi-sentence generation, and the increased capacity of the Feedback architecture is beneficial.

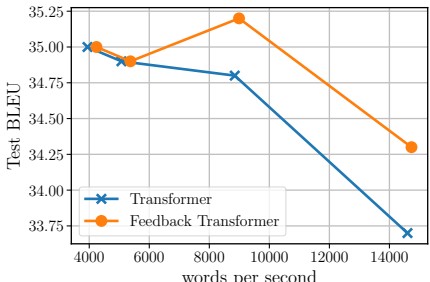
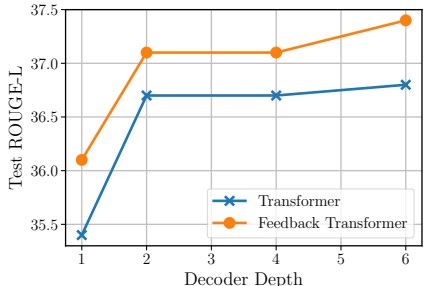

Figure 7: Results on **(left) the `IWSLT De-En` dataset,** and **(right) Summarization on `CNN-Dailymail`,** test set ROUGE-L for varying decoder depths.

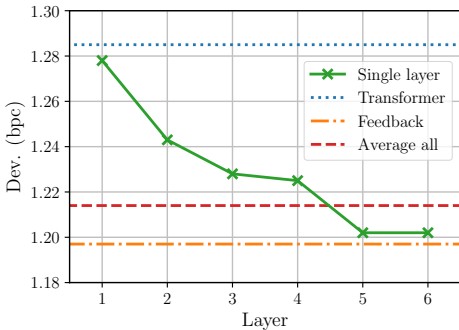

Figure 8: Ablation results on `char-PTB`: instead of a weighted sum of all layers as Feedback memory, only a single layer is used as memory for all layers. We also include a setting where the average of all layers is used.

## 6.4 ABLATION STUDIES ON LANGUAGE MODELS

We investigate which layer of a model has the best representation to be used as a Feedback memory. In Feedback Transformers, a weighted sum of all layers is used as the memory, and feeds to all layers. An alternative approach is to manually select one of the layers as the memory and let all layers attend to it. In Figure 8, we explore this approach, using the same 6-layer `char-PTB` models as Section 4.3.2 (*top-only* memory there corresponds to using the last 6th layer as memory). We can see that representations from higher layers work better as memory, confirming our assumption of the importance of higher level representations. Simply averaging all layers together works reasonably well as well. Interestingly, when all layer attend to the first layer output, it works as good as the standard Transformer. The weighted sum approach matches the best performance because it can adopt to select any of the layers.

Here we study how different techniques affect the model performance on `WikiText-103`. The results shown in Table 5 indicate:

- Pre-normalization combined with higher learning rates helps the performance, particularly for the standard Transformer.
- Increasing the context size with adaptive span further improves the performance for both models.
- The technique of increasing the BPTT length during training for efficiency does not affect the final performance.
- The gap between two model is consistent along those variations.

Next, we examine the effect of the model depth on performance on `char-PTB` and `WikiText-103` This time, we keep the total number of parameters constant and only vary the number of layers to

| Model | Pre-norm + higher LR | Adapt. span | Increase BPTT | dev ppl |
|---|---|---|---|---|
| Transformer | no | no | no | 22.9 |
| Transformer | no | no | yes | 22.9 |
| Transformer | yes | no | yes | 21.0 |
| Transformer | yes | yes | no | 20.6 |
| Feedback | no | no | no | 19.7 |
| Feedback | no | no | yes | 19.9 |
| Feedback | yes | no | yes | 19.6 |
| Feedback | yes | yes | yes | 19.0 |

Table 5: Ablation on `WikiText-103` of various modeing choices. Results are shown without finetuning.

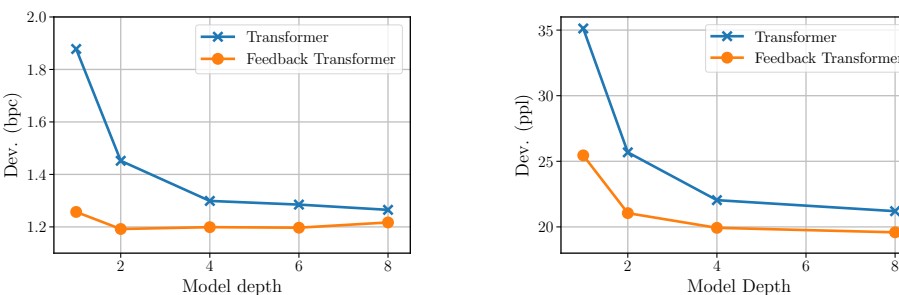

Figure 9: The performance on **(left)** `char-PTB` and **(right)** `Wikitext-103` as a function of the model depth. The number of parameters is kept constant by increasing the width.

isolate the effect of depth. This is achieved by proportionally increasing the head dimension and the ReLU layer size when we decrease the number of layers. The results in Figure 9 demonstrate that for the standard Transformer improves as the depth increase. In contrast, the Feedback architecture is much robust reduced depth, even achieving the best performance on `char-PTB` with only two layers.

# 7 ADDITIONAL IMPLEMENTATION DETAILS

## 7.1 RANDOM WALK TASK DETAILS

We provide additional details for the random walk toy task we explore. The agent starts at a fixed position of a $8 \times 8$ grid. Available actions are 1) move one step forward, 2) turn left and 3) turn right. At every time step, the agent randomly picks on of the three actions and executes it. An action would be ignored if it can't be executed like going out of the grid. After 100 actions, the agent is reset back to the initial position.

The input to the model is a sequence of actions taken by the agent, and a special symbol if there was a reset. The output is a sequence of location symbols corresponding to the agent's location after each action. We generate 10k training episodes, totalling 1M tokens.

We use the same setup as our language modeling experiments, except now the model predicts separate output tokens rather than a next token. We concatenate all the episodes and feed them to the model as a single sequence. The training is done with the negative-log-likelihood loss. See Table 8 for the hyperparameters used in the experiment. The attention span is set to 100, so that the models can attend to all the information they needs to solve the task.

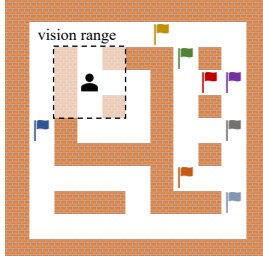 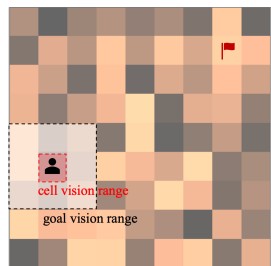

Figure 10: **(left) Maze Navigation** task and **(right) Water Maze** task.

```
x = 1 ; print x ; x ++ ; print x ; z = 8 ; print z ; print z ; x -- ; if x > z : z -- ; z ++ ;
print z ; print x ; print x ; if z < x : z ++ ; x ++ ; z -- ; x -- ; if z > x : z -- ; z ++ ;
if x > z : z ++ ; if z < 5 : y = 7 ; print x ; if x > z : z ++ ; x ++ ; y = 7 ; if x > 10 : x
-- ; y -- ; x ++ ; z ++ ; print z ; y -- ; print x ; print x ; z ++ ; y ++ ; y ++ ; if z < 3 :
y ++ ; if x > 4 : x ++ ; z -- ; x -- ; x -- ; print x ; y ++ ; z ++ ; y -- ; if x > z : z -- ;
x ++ ; z -- ; print x ; z ++ ; print y ; y ++ ; y -- ; x -- ; print x ; y ++ ; print y ; y --
; if z < x : x ++ ; if z > 4 : y -- ; z -- ; x ++ ; if y < x : y ++ ; print y ; print z ; z --
; y -- ; x ++ ; y -- ; y ++ ; if y > 3 : z -- ; y ++ ; if z < 10 : z ++ ; z ++ ; y -- ; z ++ ;
print z ; x -- ; y -- ; x -- ; x ++ ; if x < 4 : y -- ; print y ; print z ; if z > x : y -- ;
print z ; if y < x : x -- ; print x ; print z ; if x < 4 : z -- ; if z < y : z ++ ; z -- ; x --
; print x ; if z < x : y ++ ; print x ; print z ; y -- ; if z < 6 : x ++ ; z -- ; END
```

Table 6: An example program from the algorithmic task with 3 variables.

## 7.2 MAZE NAVIGATION DETAILS

We generate random $9 \times 9$ mazes using Kruskal's algorithm. Dead ends are eliminated by randomly removing one of the blocks surrounding them. We randomly place 8 target objects with different colors as shown in Figure 10 (left). The agent is given a randomly selected color as a target. If the agent manages to reach the correct target, it gets a reward of $+1$ and a new target color is sampled. An episode ends after 200 steps. The observation includes the $3 \times 3$ area around the agent and target color.

We train 2-layer Transformers with a hidden size 256 and 4 heads. We set the BPTT to 100 and the batch size to 1024. The reward discount rate is 0.99. The attention span is 200 so the agent can keep an entire episode in memory. All agents were trained using A2C with Adam with a learning rate of 0.0003 and a entropy cost of 0.0005. For the easy version of the task, we use RMSprop with a batch size of 128 and a learning rate of 0.0003. The RMSProp epsilon regularization parameter is set to 0.01 The LSTM model is a 3-layer LSTM with a hidden size of 256.

## 7.3 WATER MAZE DETAILS

The water maze task we designed is depicted visually in Figure 10 (right). The grid size is $15 \times 15$. To help exploration, the agent can see if the goal is within a $3 \times 3$ area around it. An episode ends after 200 steps. We train for 500M steps (2.5M episodes). We use 2-layer Transformers with hidden size of 64 and 1 head. The attention span is 200 so the agent can put an entire episode in memory.

All agents where trained using A2C with RMSprop with entropy cost of 0.0001, RMSProp epsilon regularisation parameter of 0.01, batch size of 64, and BPTT 200. Feedback Transformer and Transformer baseline were trained with a learning rate of 0.0003. LSTM model is a 2-layer LSTM with hidden size of 64. For LSTM model we used a learning rate of 0.0004.

## 7.4 ALGORITHMIC TASK DETAILS

In this task, each program consists of 100 simple statements that should be sequentially executed. The available statement types are:

1. **Initialization.** Assign an initial value to a variable like x=3. A variable can only be initialized once in each program.

| Hyperparameter | Summarization | WMT En-De | IWSLT De-En |
|---|---|---|---|
| Encoder Layers | 6 | 6 | 6 |
| Decoder Layers | 6 | 6 | 6 |
| FFN Size | 2048 | 4096 | 1024 |
| Attention Heads | 8 | 16 | 4 |
| Dropout | 0.3 | 0.3 | 0.3 |
| Hidden Size | 512 | 1024 | 512 |
| Learning Rate | 0.0005 | 0.001 | 0.0005 |

Table 7: Hyperparamers for sequence to sequence experiments.

| Hyperparameter | Random Walk Algorithmic | char-PTB | Enwik8 | WikiText-103 small | WikiText-103 large |
|---|---|---|---|---|---|
| Layers | 4 | 6 | 12 | 4 | 8 |
| Hidden size ($d$) | 256 | 384 | 512 | 512 | 1024 |
| FF size | $4d$ | $4d$ | $8d$ | $8d$ | $4d$ |
| Head count ($h$) | 4 | 4 | 8 | 8 | 8 |
| Head dim | $d/h$ | $d/h$ | $2d/h$ | $2d/h$ | $d/h$ |
| Attention span | 100 | 512 | 8192* | 512 | 512, 2048* |
| Dropout rate | 0.2 | 0.5 | 0.5 | 0.1 | 0.3 |
| Embed. dropout | - | - | - | 0.1 | 0.2 |
| BPTT len ($M$) | 64 | 128 | 128 | 256 | 256 |
| Batch size ($B$) | 512 | 2048 | 1024 | 512 | 512 |
| Learning rate | 0.0001 | 0.0015 | 0.0015 | 0.0007 | 0.0007 |
| Gradient clip | 0.1 | 1.0 | 0.1 | 0.1 | 0.1 |
| LR warm-up steps | 1k | 1k | 8k | 8k | 8k |
| Parameters | 3.2M | 10.7M | 77M | 44M | 139M |

Table 8: Hyperparamers for language modeling experiments. Here * indicates the adaptive span.

2. **Increment and decrement.** Increment or decrement a variable value by 1, like `x++` or `y--`.

3. **Print.** Output the value of a certain variable like `print(y)`. Only this statement requires model to make a prediction.

4. **Conditional.** Execute the nested statement only if a variable has a certain value, e.g., `if x==4: y--`. Note that conditional and print statements cannot be nested.

A program is generated by randomly choosing a statement one after another, but with the following conditions: a variable must be initialized before being used, and a variable value have to between 1 and 10. The training data contains 10k such programs concatenated with a special separator keyword. We generate two version the data with 3 and 5 different variables in them. An example program is shown in Table 6. We used the same hyperparameters as the random walk task as show in Table 8.

### 7.5 MACHINE TRANSLATION AND SUMMARIZATION

We detail the hyperparameters in Table 7. Summarization experiments are done with the Transformer base architecture size and WMT En-De experiments are done with the Transformer big architecture size. As IWSLT De-En is a smaller dataset, we use a smaller model. For all sequence to sequence experiments, only the decoder is modified to have the Feedback Transformer architecture.

### 7.6 LANGUAGE MODELING

In the language modeling experiments, we added several improvements on top of the original Transformer Vaswani et al. (2017) to better adapt to unbounded sequences:

- **Hidden representation caching Dai et al. (2019):** Since the input to the model is an unbounded sequence and the model needs to process it in small blocks, hidden representations from previous blocks are kept in cache so that any token in the current block will the same context length regardless of its position in the block.
- **Relative position embedding Shaw et al. (2018):** Relative position embeddings allow each token in a block to be processed in the same way regardless of its absolute position in the block. We found that adding shared embeddings to key vectors at every layer to be effective.
- **Adaptive attention span Sukhbaatar et al. (2019)** Language modeling requires a model to have a very long attention span, which is computationally expensive. The adaptive span mechanism allows each attention head to learn different attention spans for efficiency.
- **Pre-normalization Child et al. (2019)**: We observed that pre-normalization makes training more stable for Transformers, which allowed us to use larger batch sizes for better parallelization.

Dropouts are applied to attention and ReLU activations. In `WikiText-103` models, additional dropouts are added to the embedding layer output and the last sublayer output.

In Table 8, we present the hyperparameter values used for our experiments. We use the same hyperparameters for both Transformers and Feedback Transformers, and optimize them with Adam. The final performances are obtained by finetuning the models with a 10x smaller learning rate.

**Details on the `char-PTB` experiments** We trained the models for 15k updates (or earlier if the validation loss stops decreasing), and funetined them for 1k steps. We varied the depth of the models while keeping the number of parameters constant. This is achieved by changing the FF size and the head dimension inverse proportionally to the depth.

**Details on the `enwik8` experiments** We used an adaptive span limited to 8192 tokens with a loss of 0.0000005. The training is done for 100k updates and another 10k steps is used for finetuning. The warming up BPTT length is used for speeding up the training, where the BPTT length is decreased to 64 for the first half of the training.

**Details for Training on `WikiText-103`** We employed the adaptive input Baevski & Auli (2019) and the adaptive softmax Grave et al. (2017) techniques for reducing the number of parameters within word embeddings. The models are trained for 200k steps and the finetuned for additional 10k steps.

While most of the models have a fixed attention span of 512, the best performance is achieved by extending the attention span to 2048 with adaptive span loss 0.00001.

After training our models, we noticed that our tokenization method differed from others by omitting end-of-line (EOL) symbols. Since our dictionary already contained the EOL token, we were able finetune our trained models on the data with EOL tokens, rather than training them from scratch. This change alone brought about 1ppl improvement.

