# OpenReview forum: "Addressing Some Limitations of Transformers with Feedback Memory"
_ICLR.cc/2021/Conference — Reject_

### Official Review · AnonReviewer4 · 2020-10-27
**Review summary**

**Rating:** 5
**Confidence:** 5

**Review:**

The authors try to identify several problems in the Transformer model and modify the model architecture.

Major:

1. The authors argue that for any position k and layer l, the standard Transformer can only access previous positions (<k) and lower layers (<l). Instead, the authors propose to leverage >l layers for <k positions. First, apparently, compared to standard Transformer, the training of this model (teacher forcing setting) is much slower as the computation of any positions requires the whole forward outputs of all previous positions (Standard Transformer run positions together (in parallel)). The author should demonstrate the training efficiency of the proposed model.

2. As far as I know, for other architectures, such as deep RNN(LSTM) or convseq, as the same as Transformer, the computations of any position k and layer l only access to previous positions (<k) and lower layers (<l). Therefore, I think the authors should discuss the problems in those model architectures and test their proposal in more settings.

3. The strong requirement of a belief state in a model architecture is not convincing evidence to me. Or I can also view the ffn outputs at any (position, layer) as a virtual belief state. I also have difficulty understanding the arguments in the 'Maintaining a Belief State' paragraph for concrete reasons. I hope the author could pay more attention to describing the motivation behind.

4. The authors argue that the proposed model is memory efficient than the standard Transformer, but this seems to be not a fair comparison. They share the key and values across different layers as in Albert, Universal Transformer, and DEQ, but fail to connect to these previous works. This trick cannot be viewed as a contribution to the paper.

5. Regarding experiments and comparisons.
a. If you need to highlight long memory tasks (table 1),  please include the Transformer-XL, sparse Transformer into the comparison, which are very typical baselines in this scenario.
b. For the experiment in 4.2.1, the single decoding layer setting, what is the difference between your model and Transformer? In such a setting, both model access to all previous states. Where does the benefit come from?
c. In section 4.2.1, in the main body, you write the performance of your model (12-layer encoder + 12-layer decoder) is 29.0. But in Table 2, you write the performance of your model is 29.5, but the baseline models are only (6-layer encoder + 6-layer decoder)
d. If you need to highlight the fast decoding, please include the non-autoregressive models and linear transformers as baselines.


Minor:

1. In the 2nd paragraph, the 'feedforward nature' is not clear.

Overall:

The general problem that the authors want to solve is not very clear or very well-motivated. The experimental comparisons and baselines are not adequate. There is much room for better paper writing and presentation.

---

> ### Author Response · Authors · 2020-11-17
> **review response**
>
> Thanks for your review! We respectfully disagree with several points raised in the review, such as:
>
> - *“For the experiment in 4.2.1, the single decoding layer setting, what is the difference between your model and Transformer? In such a setting, both model access to all previous states.”* → A standard transformer would only have access to token embeddings when computing output representations of layer 1.
> - *“such as deep RNN(LSTM) or convseq, as the same as Transformer, the computations of any position k and layer l only access to previous positions (<k) and lower layers (<l)”* → Actually, RNNs can only access their own layer (let’s call it L) and the layer immediately below (L-1).
> - *“They share the key and values across different layers as in Albert, Universal Transformer, and DEQ, but fail to connect to these previous works.”* → This is incorrect. We share keys and values (**activations**), while these listed models share **parameters** across layers.
> - *“the part about feedforward nature is not clear”* → Transformers are Feedforward networks, and RNNs are not.
>
> Respectfully, we would like to clarify a few points that could help make our contributions more clear.
>
> **re: How Feedforward Networks and RNNs operate-** A feedforward neural network (re: “the part about feedforward nature is not clear”) is a neural network where the connections between neurons only move in one direction. MLPs, convolutions, and Transformers are Feedforward, but RNNs and LSTMs are not. Your statement about RNNs accessing all previous positions and all lower layers is sadly not true. RNNs can only access their own layer (let’s call it L) and the layer immediately below (L-1). Thus, the Feedback Transformer architecture we propose is distinct from RNNs. In Figure 5, you can see an ablation where if Feedback Transformer was basically an RNN, it does not work very well. So it’s important to be able to leverage computation across all layers.
>
> **re: Comparisons to Other Transformers regarding key/value sharing -** This is incorrect. In our work, we propose to share keys and values which are activations of the model, while ALBERT or Universal Transformers propose to share parameters across layers. For many (most?) use cases, activations actually use more memory and compute than parameters.
>
> **re: what is Belief States-** We want to clarify that we don’t mean “belief state” as an abstract concept, but belief state as in - what is the model’s current internal representation. Having a belief state is absolutely critical. For example, in a task about moving around a maze, the model needs to understand where it is in the maze to decide where to move next. This is also true in our new algorithmic task - without this ability, models cannot update information about the world. You can see the clear performance difference between standard Transformers and Feedback Transformers.
>
> **Detailed questions about Experiments and Comparisons:**
>
> **re: compare to Transformer XL and sparse Transformer**  Note that we compare to Transformer XL in real tasks, and compare to models with better results compared to both Transformer-XL and Sparse Transformer in Tables 3 and 4. Tables 3 and 4 evaluate on models with much longer sequences than Table 1, so we feel we have included the comparisons where it is most relevant. . Further, Transformers and Transformer-XL are equivalent when feeding the whole sequence at once (up to the position encoding, though we also include relative position embeddings as used in Transformer-XL). Recent analysis on various Long-Range Transformers (https://arxiv.org/pdf/2011.04006.pdf) indicates that our Transformer baseline is very strong.
>
> **re:  What is the difference between Feedback and Transformer in single layer setting?** This is also incorrect. In a one layer model, our architecture would access the output representation of layer 1 of positions <k when computing the output representation at position k of layer 1. On the other hand, a standard transformer would only have access to token embeddings when computing output representations of layer 1.
>
> **re: fast decoding compared to non-autoregressive and linear Transformers** - Kasai et al. (2020) show that a deep encoder with shallow decoder is a very strong baseline for efficient NMT systems, competitive with current non-autoregressive models. This is why we used this as a baseline, and not NAT models. We show that our architecture can even improve over this, leading to a strong model for efficient NMT.
>
> J. Kasai, N. Pappas, H. Peng, J. Cross, N. A. Smith. Deep Encoder, Shallow Decoder: Reevaluating the Speed-Quality Tradeoff in Machine Translation.

---

> ### Author Response · Authors · 2020-11-19
> **any additional questions?**
>
> Let us know if there's anything that's unclear or any additional questions that we can answer before the discussion period ends.

---

> > ### Comment · AnonReviewer4 · 2020-11-24
> > **Response from R4**
> >
> > I thank the authors very much for carefully addressing my concern, and I am sorry for the late response as I am quite busy this a few days. Regarding your response:
> >
> > * Comparison between Universal Transformer and Albert:
> >
> > You are correct. I hope you can make this clear in a new version of the paper.
> >
> > * Regarding the term feedforward.
> >
> > I could understand what it means according to your feedback and accept it. But I hope you can replace the term 'feedforward' with some other term to make little confusion: I am not sure if all people in the community view RNN not as feedforward.
> >
> > * The difference between feedback network and a single-layer Transformer
> >
> > Yes. You are correct. But again, I hope you explicitly mention this difference in the paper.
> >
> > * Regarding NAT baselines.
> >
> > Yes. You convince me that your baseline is reasonable.
> >
> > * Regarding whether RNN/convseq can access previous positions and lower layers.
> >
> > I defend it. It is well-known that the RNN can leverage sequence information, although, at each position, the RNN cell can only access the cell state and current input. Such an argument still holds for deep RNN. For deep RNN, a cell in one layer can only access the layer exactly below it, which contains the information of all lower layers. Not to mention that sometimes you can use the residual connection, which creates a shortcut to lower layers.
> >
> > That is why I ask if your whole story holds, you should apply your method to RNN and convseq and check whether there are still some gains since both of them cannot access upper-layer information (using a similar argument).
> >
> > The authors did a quite good job addressing some of my concerns, and I raise my score. However, I think the key argument ("The representation at a given layer should access representations from upper layers") is still not well-addressed.
> >
> > I would also like to discuss this with other reviewers.

---

> > > ### Author Response · Authors · 2020-11-24
> > > **Thanks, and response to response**
> > >
> > > We thank the reviewer for the detailed review and discussion, we appreciate it. Below, we provide additional details to help clarify our contribution, and we will make sure that these clarifications (as well as the others above) will be clear in the final version of our paper. Thanks for the consideration, and thanks for raising your score!
> > >
> > > **re RNN + Conv**
> > >
> > > We believe that *applying your method to RNN and convseq* is out of the scope of our paper, and should not be basis for rejection, as we did not make the claim that our method generalizes to other architectures. Our proposed method is an architecture change specific to Transformers, which became a widely used model, especially for sequential data. We address a specific limitations of such models (lack of recurrence, see discussion below), and provide evidence that this change has benefits. Besides, as mentioned in the related work, there is already a paper [1] showing that feedback connections work in RNNs too.
> > >
> > > **re Difference between Feedforward and Recurrent**
> > >
> > > We agree that (deep) RNNs, ConvNets or Transformers cannot access representations of upper layers when computing the representation of a given layer. However, this was not our argument, which needs clarification. What we meant is that there is a fundamental difference between recurrent models (RNNs, LSTMs, Feedback Transformers) and feedforward models (ConvNets, Transformers). Feedforward as terminology perhaps is slightly under-used, but it is well accepted that RNNs are not feedforward networks (see the introduction of Chapter 6 of the Deep Learning Textbook, where explicitly it is mentioned that when networks include feedback connections - like Feedback Transformer or RNNs, they are no longer feedforward).
> > >
> > > Specifically, the longest computational path in (deep) RNN and feedback Transformers is proportional to the length of the input sequence plus the number of layers. However, in feedforward models, it is proportional to the number of layers only. The fact that computational paths depend on the length of the data is important, and is highlighted in our algorithmic task where very long computational paths are required (please see our responses to AnonReviewer2 for more details on computation in the algorithmic task). Another motivation for this is the theoretical analysis of [2], showing that standard Transformers cannot recognize certain formal languages, unless the number of layers or heads increases with input length (while fixed size RNNs can recognize these languages).  Finally, a last evidence of this is the gated convolutional language modeling paper: as the computation in strictly feedforward models is proportional to the number of layers, it required 14 convolutional blocks (each a bottleneck convolution of three layers) to be competitive with the 2 layer large LSTM [3]. This is why it is important that we have provided a straightforward modification to Transformers that provides additional capacity, and is why our model can have stronger results even though it is shallower and smaller.
> > >
> > >
> > > [1] Chung et.al., Gated Feedback Recurrent Neural Networks, 2015. https://arxiv.org/abs/1502.02367
> > >
> > > [2] Hahn, Theoretical Limitations of Self-Attention in Neural Sequence Models, 2020, TACL.
> > >
> > > [3] Dauphin et al, Language Modeling with Gated Convolutional Networks, 2018. https://arxiv.org/pdf/1612.08083.pdf

---

### Official Review · AnonReviewer3 · 2020-10-28

**Rating:** 6
**Confidence:** 3

**Review:**

### Summary
This paper modifies transformers with feedback memory. Specifically, for each timestep, it merges hidden representations of all layers into a high-level single vector and stores it in memory. For the current timestep, it attends past memory vectors. The authors claim that in this way, low layers of the current timestep can utilize high-level representations of past timesteps. The authors show that the proposed models with shallow layers can achieve stronger performance than comparable transformers. However, it seems that the models need a much longer time to train.


### Strengths
* With feedback memory, the modified can speedup decoding (autoregressive generation) as shown in Figure 4.
* Since the proposed model can directly utilize previous high-level representations, it just needs a small size and shallow layers to achieve comparable performance as shown in Table 3 and Table 4.

### Weaknesses and Questions
* Training time and inference speed are important for such practical models. It is better to complement these to Table 1/2/3/4. It seems that the proposed needs to take a much longer time to train as the authors mentioned it in a sentence on page 8. The authors can give more results and discussions so that future users can know whether to choose transformers with feedback memory according to their situations.
* (optional) In Table 3/4, how about feedback transformer that keeps the same number of layers and similar parameters as Trans-XL. Transformers usually can achieve better performance when the number of layers increases.  It is just an optional discussion as feedback transformers seem to need much time to train and the rebuttal time is limited.

---

> ### Author Response · Authors · 2020-11-17
> **review response**
>
> Thanks for your review! We’ve included additional analysis to respond to your questions, and included several additional experiments overall in the paper to strengthen our work. Please let us know if you have additional questions.
>
> **re: training and inference time analysis**
> We include a table here about the training and validation speed of the Feedback Transformer compared to the standard Transformer for language modeling, translation, and reinforcement learning. For language modeling, we measure on Wikitext-103 (we compare a 8-layer Transformer against a 4-layer Feedback model of the same size. Both models have a fixed attention span of 512, and trained on 32GPUs. With the Feedback model, we’re able to fit 2x larger batches in GPU memory. The inference is done with 1GPU). For translation, we measure on WMT En-De with 6 layer encoder and 2 layer decoder (reporting training WPS on 8 GPU and inference WPS on 1 GPU). For RL, we report the training frame-per-second (FPS) on the maze navigation task (using 20 CPU cores and 1 GPU). We will add this table into the main paper.
>
> |Task |Model | Training WPS | Inference WPS `|
> | --- | --- | --- | --- |
> |LM (wiki103) |Transformer | 296K | 592 |
> |LM (wiki103) |Feedback |84.K | 2176 |
> |Translation |Transformer | 280K | 3190 |
> |Translation | Feedback |126K | 5410 |
> |RL Maze | Transformer | 22.3K | --- |
> |RL Maze |Feedback | 22.3K | --- |
>
> For Encoder-Decoder tasks, the Feedback Transformer is slower than the standard Transformer, but is faster at Inference as it uses less memory and can thus generate translations with larger batch sizes. For Language modeling, the Feedback Transformer is about 3x slower to train, much faster at inference due to reduced memory cost (from sharing key-values) and reduced depth. For reinforcement learning tasks, the training must be online as well, so the Transformer and Feedback Transformer have the same speed.
>
> **re: larger feedback models**
> We will explore training larger Feedback Transformer models. For datasets like Wikitext-103 and WMT En-De, it is mainly a challenge of regularization, so we anticipate needing to tune the dropout parameters. We'll add additional results in the final version of the paper based on this exploration (as you mention, the training time is a challenge in the short-ish rebuttal period).
>
> However, we want to emphasize that the main contribution of the paper is a straightforward solution to two major limitations of the Transformer architecture (limited access to higher level representations and inability to maintain state). By resolving these limitations, it's possible to have a model that is smaller and shallower perform just as well. We're happy to conduct additional experiments, but the point isn't really chasing state of the art performance by sweeping more, but a simple way to not have these limitations in a Transformer.

---

> ### Author Response · Authors · 2020-11-19
> **any additional questions?**
>
> Let us know if there's anything that's unclear or any additional questions that we can answer before the discussion period ends.

---

### Official Review · AnonReviewer2 · 2020-10-28
**Good empirical results on LM and RL; lack of more detailed efficiency & large-scale RL validations**

**Rating:** 6
**Confidence:** 5

**Review:**

> Summary: This paper proposes some changes to the classical Transformer architecture to address its major limitations, such as limited access to higher-level representations. It specifically introduces recurrence to the Transformer architecture by feeding the activations of all previous time steps to a later time step (in the form of self-attention). Empirical results on language modeling and small-scale RL tasks seem to suggest the usefulness of doing do.

--------------------

Post-rebuttal thoughts:

See the comment block below.

--------------------

Overall:

I found this paper interesting and relatively easy to follow. The idea is simple, and seems useful, although I do find some arguments handwavy and not quite convincing (e.g., the "maintaining a belief state" one). It is unclear to me how exactly the efficiency compares, though the authors did report the # of days on WikiText-103 (see my detailed question below). I overall think that this could be a good architectural improvement on the condition that the authors provide more details.

Pros:

1. Simple idea and the flow of the paper is easy to follow.
2. An extensive set of experiments to verify both the usefulness of the Feedback Transformer and the limitations that the authors hypothesize to be true for transformers.

Cons:

1. The introduction of sequential-ness to Transformer is good but obviously would slow things down especially as the sequence gets longer. The authors reported on this very briefly, but I think it is an important enough aspect to warrant more analysis.
2. Lack of certain ablative settings in the experiments (which is unavoidable in a certain sense, given that the paper proposes various changes to the architecture).

----------------------------------------------

Additional comments and questions:

1. The core of the hypothesis on the value of high-level representation feedback is the autoregressiveness, is this correct? As the paper claims, typical Transformers are restricted from "taking full advantage of the input's sequential property" because they can't access the higher-level representations of previous time steps. I have two questions in this respect, and wonder if the authors have verified this (if not, I think you probably should (?)):
    1) Would you expect a "feedback LSTM" to work better than an LSTM as well? In other words, an LSTM that when computing $h_t^{(l)}$ of time $t$ at layer $l$, uses $h_{<t}^{(L)}$ just like in the Feedback Transformer?
    2) In pixel sequences like CIFAR-10 density modeling (e.g., see Sparse Transformer by Child et al. 2019), where the autoregressiveness is not rather obvious (e.g., you can do column-based or row-based, or even Hilbert curves), does Feedback Transformer still help? If so, then it means higher-level representation is not exactly a "temporal" phenomenon, because there's nothing in pixels that's temporal...

2. Regarding the sharing of keys and values in a Feedback Transformer--- is the motivation for this just to speed up the architecture? How well does Feedback Transformer perform without this sharing, and how slow would it be?

3. I'm confused about the "maintaining a belief state" paragraph. The authors claim that Transformers are limited by "only a fixed number of transformations can be applied to its internal states". But aren't those internal states already aggregated by lower levels? Why might more transformations be better? Can't one simply increase the number of layers of a Transformer? I also don't see the logical connection between this claim and the end of this paragraph: "This means Transformers cannot maintain an internal state for a long time if it has to be frequently updated". Can the authors clarify on this part?

4. In cases especially like NMT, where decoders are trained in parallel (because at training time, the decoder is trained just like an LM) and used for inference in sequence (at test time, it generates tokens one by one), wouldn't it make more sense to pre-train a classical Transformer (with no feedback memory) and then directly use, or probably with slight fine-tuning, the feedback version of it at inference? Is this possible?

5. One of the most important thing that I believe the current version is missing is a more comprehensive analysis of the efficiency, which seems to be an important drawback (if any). I noticed that the authors claim that key-value sharing compensated for the loss on parallelism--- but by how much exactly? Specifically, I'd appreciate if the authors can provide an analysis of at least some of the following:
    1) How many GPUs (and what sort of GPU) did you use to train your models, e.g., for WikiText-103 and for char-PTB? Did you use the same setting for the classical Transformer? (The 1.2 vs. 3.5 days on WikiText-103 is still a large gap, almost 3x slower...)
    2) How does the efficiency of the training (not in terms of days of training, but ms per batch) scale as you use longer and longer sequences? I'm asking because I noticed in Table 7 that these sequence lengths are still pretty short; e.g., I believe SOTA char-PTB uses length > 256 and WikiText-103 uses length > 1024 at inference. Does Feedback Transformer further improve when you use longer sequences?
    3) If the "high-level representation" of Transformers is indeed a major limitation, does a deeper (but still the same # of parameters, so probably smaller hidden dimensionality) Transformer perform better, because it can have "more updates" to its internal state? Or maybe a weight-sharing Transformer? How does the efficiency vs. performance compare in these cases?

6. Did you train all of the Feedback Transformers from scratch (i.e., `train_step`=0), or did you warm-up/pretrain the models?

7. Have you ever tried non-toy-scale RL tasks? I think this proposed architecture would be very useful in these very sequential settings (e.g., in robotics, where the data stream actually has temporal dimensions), and these large-scale RL tasks (e.g., Doom FPS game, etc.) could make the paper even stronger.

8. I'd suggest expanding Tables 3 and 4--- there are plenty of prior works that evaluated on these two datasets and it'd be worth it to cite them to compare. In addition, for Table 3, does increasing parameters further improve the performance? It is impressive that you can achieve the same level of result as Transformer-XL with only half of the parameters, which seems to suggest there's still room for improvement?

9. Section 4.3: "we first" ---> "first"

---------------------------------

Overall, I think this paper presents relatively solid results, but there are some key ablative settings, efficiency details & analysis, and large-scale RL results that are missing. I'm putting a 5 on this paper for now, but I look forward to the authors' response and am happy to adjust my score positively once my questions are further clarified.

---

> ### Author Response · Authors · 2020-11-17
> **review response, part 1**
>
> Thanks for your comprehensive questions and detailed reading of the paper! Below, we’ve responded to each of your questions and included several new experiments.
>
> **re: Speed as Sequence gets Longer**
> For long sequences, we do not process from start to end. Instead, for both standard Transformer and Feedback Transformer, we process the sequence in blocks of fixed size in the same way as TransformerXL, so the speed difference between the two remains constant, regardless of the data length. With regards to longer attention spans, it will not slow down Feedback Transformer more than standard Transformer because both process attention spans in parallel. At inference, both decode with the same speed as decoding proceeds token-by-token no matter what.
>
> **re: Ablations**
> We have an ablation study in Sec 4.3.2. In addition, we also have added a number of new experiments in our review response. Let us know if you have specific suggestions!
>
> **re: feedback LSTM and temporal phenomenon**
> For Feedback LSTM, there is an existing work exploring it and shown improvements [1] (we mentioned this in the related work section, but we will make it clearer). For your second question, yes, it doesn't have to be temporal only. As long as computation is autoregressive in a certain dimension (e.g., temporal or spatial), the Feedback mechanism can be applied.
>
> [1] Chung et.al., Gated Feedback Recurrent Neural Networks, 2015 https://arxiv.org/abs/1502.02367
>
> **re: motivation to share keys and values**
> The motivation is exactly as you state, for efficiency. Note that the standard Transformer cannot share keys and values (because they are computed from different representations at different layers), so this is only possible for the Feedback Transformer. Sharing improves the speed around 3x for training and also reduces the memory required by the model, which contributes to faster inference speed as well (the memory can instead be used to increase batch size). In our ablation experiments, we did not find performance difference between sharing and not sharing in the Feedback Transformer.
>
> **re: maintaining belief state and model depth**
> Good question. One way to think about it is - how many nonlinear functions can a model apply on any state? The Feedback Transformer has recursive computation, so it can continuously iterate. The standard Transformer can use each layer to change its internal state. Yes, a deeper Transformer can manipulate its internal state more, but this has clear limitations. One limitation is with regard to sequence length. Can we really scale model depth with sequence length? What if the model needs to change internal state a large number of times to carefully track something? This outstrips the rate at which we can grow the # of layers and train deep models stably.
>
> This is illustrated in the existing maze task, but we have added an algorithmic task involving code execution to further illustrate this idea. It's added in the paper in Section 4.1.2. The model gets some variables and the state of those variables is constantly being updated. You can see that the deeper Transformer is better than the shallow Transformer, but LSTM does much better than both, and Feedback easily does the best. This is because the Transformer needs to constantly track the variable values, and quickly runs out of capacity. We will put this data online for others to try as well.
>
> **re: finetuning from standard transformer**
> We have explored previously training a standard Transformer model and then finetuning into the Feedback architecture - it is definitely possible. We add a table below indicating the performance of such a finetuning strategy.
>
> |Model |Performance|
> | --- | --- |
> |Transformer Baseline | 21.2 |
> |Feedback Transformer | 19.7 |
> |Transformer Finetuned to Feedback   | 19.8 |
>
> However, since the Feedback Transformer substantially changes the way the model works (key and value vectors feeding to self-attention sublayers completely change), you cannot train with one architecture and apply another at inference time - thus, it takes time to do the finetuning and create the Feedback architecture. With the speedup from sharing keys and values, we found no real benefit from finetuning compared to training a Feedback Transformer fully from scratch. Other ways to speed up convergence still apply --- in the translation setting, for example, initializing with pretrained embeddings is totally possible, but would improve the convergence speed of the standard Transformer as well, so we do not assess.

---

> > ### Author Response · Authors · 2020-11-17
> > **review response, part 2**
> >
> > **re: analysis of efficiency**
> > **a) GPUs and efficiency analysis**
> > We provide a detailed breakdown to compare the Transformer and Feedback Transformer. We use V100 GPUs and run both models on the same infrastructure for all experiments in the paper.
> >
> > We include a table here about the training and validation speed of the Feedback Transformer compared to the standard Transformer for language modeling, translation, and reinforcement learning. For language modeling, we measure on Wikitext-103 (we compare a 8-layer Transformer against a 4-layer Feedback model of the same size. Both models have a fixed attention span of 512, and trained on 32GPUs. With the Feedback model, we’re able to fit 2x larger batches in GPU memory. The inference is done with 1GPU). For translation, we measure on WMT En-De with 6 layer encoder and 2 layer decoder (reporting training WPS on 8 GPU and inference WPS on 1 GPU). For RL, we report the training frame-per-second (FPS) on the maze navigation task (using 20 CPU cores and 1 GPU). We will add this table into the main paper.
> >
> > |Task |Model | Training WPS | Inference WPS `|
> > | --- | --- | --- | --- |
> > |LM (wiki103) |Transformer | 296K | 592 |
> > |LM (wiki103) |Feedback |84.K | 2176 |
> > |Translation |Transformer | 280K | 3190 |
> > |Translation | Feedback |126K | 5410 |
> > |RL Maze | Transformer | 22.3K | --- |
> > |RL Maze |Feedback | 22.3K | --- |
> >
> > For Encoder-Decoder tasks, the Feedback Transformer is slower than the standard Transformer, but is faster at Inference as it uses less memory and can thus generate translations with larger batch sizes. For Language modeling, the Feedback Transformer is about 3x slower to train, much faster at inference due to reduced memory cost (from sharing key-values) and reduced depth. For reinforcement learning tasks, the training must be online as well, so the Transformer and Feedback Transformer have the same speed.
> >
> > **(b) efficiency with longer sequences**
> > please see our response above, where we respond to Con #1. An important metric for good performance in long-context tasks is attention span, and we do train with very long attention spans (2048 for WikiText-103, and 8k for Enwik8). Feedback Transformer processes keys and values in parallel, thus it does not become slower with longer spans more than standard Transformer. We see both models improve with longer spans, but Feedback always outperform. In fact, a Feedback Transformer reached a new SOTA on Enwik8 which requires a much longer context size than WikiText-103.
> >
> > **(c) are deeper transformers more effective? what about sharing parameters?**
> > In the Appendix, Figure 9, we have an analysis of making models deeper while keeping the parameter count fixed (exactly as you mentioned, by reducing the width of each layer). We compare a Feedback Transformer and a standard Transformer in this setting on Wikitext-103 and Penn Treebank. You can see that Feedback Transformer still does better, even if the model is deeper.
> >
> > Regarding weight sharing Transformer, existing work has explored this already - such as Universal Transformer or ALBERT, but still retain these limitations.
> >
> > **re: train from scratch v. warm up**
> > see our response to your question (4) above. We tried warm-up/pretrain as you mention and include the additional results.
> >
> > **re: larger scale RL tasks like doom FPS game**
> > We are interested in the future in scaling on larger RL tasks and definitely consider this future work.
> >
> > **re: expand table 4 and 5**
> > Sure, we can add many additional citations for each task. However, we want to emphasize that the main contribution of the paper is a straightforward solution to two major limitations of the Transformer architecture (limited access to higher level representations and inability to maintain state). By resolving these limitations, it's possible to have a model that is smaller and shallower perform just as well. We're happy to add many references (especially with the additional space in the final version), but the point isn't really chasing state of the art performance, but a simple way to not have these limitations in a Transformer.
> >
> > Regarding room for improvement, we will explore training larger Feedback Transformer models for language modeling and translation. For datasets like Wikitext-103 and WMT En-De, it is mainly a challenge of regularization, so we anticipate needing to tune the dropout parameters. We'll add additional results in the final version of the paper.
> >
> > **re: typo**
> > fixed, thanks!

---

> > ### Comment · AnonReviewer2 · 2020-11-20
> > **Additional questions**
> >
> > Thank you for your response. Some additional questions:
> >
> > 1. Why is Feedback Transformer faster than Transformer on LM inference? Should LM inference function in the same way as in training (i.e., parallel processing)? Or are you using sequential generation for LM testing evaluation? Interestingly, in MT Feedback Transformer, where the generation is **actually sequential** and the same encoder structure is used, the speed up is rather small compared to LM. I wonder what's the cause of this.
> >
> > 2. Regarding the belief state propagation thing--- sure, it might be the case that sometimes more activations are needed. But this kind of activation stacking are also known to suffer from vanishing gradients, right? If I were to guess, eventually the activations that matter are still those that only undergo a limited number of transformations (maybe more than the # of layers, as you indicated). My point is, it's hard to argue that whether the activation stacking is something that is "the more the better". In your case, in order to fully substantiate this claim, for example, it would be more useful to directly study the gradient weight. In the random walk experiment you show, the fact that in the "5 var" case 8L Transformer and LSTM perform worse than 4L Transformer could be an indication that things are not as simple as the activation count.
> >
> > 3. Interesting. I would have guessed that the way Feedback Transformer processes a sequence implies its GPU utilization might be lower than that of the Transformer, and thus the speed gap would grow linearly with the sequence length. Did you actually try this; e.g., running Transformer and Feedback Transformer (with all other conditions fixed) on seq. lengths of, say, [64, 128, 256, 512]?

---

> > > ### Author Response · Authors · 2020-11-20
> > > **response to additional questions**
> > >
> > > Thanks for your additional questions and engaged discussion! Our responses are below.
> > >
> > > **re question 1**:  For LM, yes, “inference” was in sequential generation setting, as if you were using a language model to generate. Not scoring PPL on the entire sequence at once, which would be parallel processing just like during training. The speed up mostly comes from sharing key and value computations for all layers. For MT, we investigated and believe the problem is with the batch size packing. At inference time, we didn't pack the batch size to the maximum possible for both models, which affects the Feedback model's speed because we don't take advantage of the GPU memory freed by the key-value sharing. Thanks for helping us identify the problem. We've updated the numbers now in the original table, and include the specific change below. The transformer baseline marginally improved in speed, but the Feedback Transformer improved a lot. Note that because the encoder is a standard Transformer and not Feedback (as the entire input for translation is available simultaneously, even at inference time), the speed-up is not as large as compared to language modeling, where the entire model is the Feedback architecture.
> > >
> > > |Task |Model | Training WPS | Inference WPS |
> > > | --- | --- | --- | --- |
> > > |Translation |Transformer | 280K | 3190 |
> > > |Translation | Feedback |126K | 5410 |
> > >
> > >
> > > **re question 2**: Yes, you are right that information going through many transformations will have problems like vanishing gradient. But, this more acute in RNNs because all activations have to transform at each time step, so gradient vanishing has a direct correlation with the number of time steps. In contrast, in the Feedback Transformer, information in a memory stays the in there, fixed, until it's read by attention and goes through a transformation. Thus, gradient vanishing will correlate to the actual number of transformations needed, which can be much smaller than the number of time steps. Finally, the activation count, or the number of transformations that can be stacked, in a 4L Feedback Transformer can be larger than 4. This is because an activation can go through 4 transformation at time $t$, then feedback to the first layer and go through another 4 transformations at $t+1$.
> > >
> > > **re question 3**: 1. It will depend on what do you mean by sequence length. To clarify an implementation detail for better context: we follow Transformer-XL in processing sequences as blocks. The size of each block is the backprop-through-time (BPTT) length. So even if the sequence is thousands of tokens long, the Feedback Transformer does not BPTT through all thousands of them, but is capped by the block size.
> > >
> > > If it is the length of input data, then both models process it in small blocks, so the speed gap will not grow. But if you meant the block size (i.e. it is also BPTT length), then yes, increasing it will slow down Feedback more. However, there are several things to consider:
> > >
> > > **a.** Most of our experiments do use large block sizes (256 and 512) so the setting we report is actually a setting that’s not that favorable for the Feedback Transformer. However, despite that, the Feedback Transformer still reports strong efficiency numbers in terms of training speed and generation speed (which we refer to as inference in the table).
> > >
> > > **b.** The block size always can be reduced in exchange for larger batch sizes during training, which will increase the parallelism of the Feedback Transformer. However, since the block size also determines the BPTT length, too small values can degrade performance. This is why we used large block sizes.
> > >
> > > **c.** Increasing the block size will slow down normal Transformers too because a GPU has a limited parallel computation capacity. With a larger batch size, this capacity will be filled at a smaller block size. Increasing the block size beyond this will linearly increase the compute time for the normal Transformer, as well as the Feedback Transformer. Therefore, for very large blocks sizes, both models will slow down at the same rate.

---

> > > > ### Comment · AnonReviewer2 · 2020-11-20
> > > > **More follow up questions/discussions**
> > > >
> > > > Following up on question 2, do you have a good sense of why in Table 1's "algorithmic task", a "Transformer 4L" performs best, better than a deeper Transformer and an LSTM? Doesn't this somehow suggest there is more than just the number of activation paths that play a role in modeling these "internal states"?
> > > >
> > > > And regarding larger-scale RL, have you *ever* tried anything larger than the ones shown in this paper (I know you said "future work" in the response below, but it seems like a pretty natural thing to do while researching this architecture)? E.g., even Atari, etc.? Does it seem to capture internal state change better or worse than RNNs? Is there any stability issue? Even preliminary results would be good to show at this point :-)

---

> > > > > ### Author Response · Authors · 2020-11-21
> > > > > **Response to additional questions**
> > > > >
> > > > > Thanks for all of the questions, we're excited about the engagement :)
> > > > >
> > > > > **re question 1**: To solve this task, a model has to store current variable values (e.g. x=4, y=3, z=6) in its activations, and update them according to inputs (e.g. x++). Although a model can aggregate multiple updates (e.g. x++; x--;) to a single update, it’s not possible when there is a conditional between them (e.g. x++; if x=5: y—; x--).
> > > > >
> > > > > - So why Transformers cannot do this? Let’s take an example where a value “x=4” is stored in activations at the k-th layer. When there are inputs “x++”, operation “x+1=5” cannot be done at the k-th layer because it can only see the lower layers and information “x=4” is at the k-th layer. Therefore, “x=5” can only be found at the k+1-th layer. For a L-layer model, x’s value can only be found at the last layer after L updates, and cannot perform the next update. When this happens, the model will forget x’s value and fail at the task.
> > > > >
> > > > > - Why LSTMs fails? An LSTM can store “x=4” in its recurrent activations. This activation gets updated with each input, so input “x++” can be combined with “x=4” to produce “x=5” in the same layer. In this sense, LSTMs can process unlimited such updates. However, a problem arises when there are multiple variables. Recurrent activations are all in a single vector, so multiple variable values (e.g. x=4, y=3, z=6) have to be stored there together, which is not that hard. But it’s very hard to update one the variables while keeping the others intact. It will require a robust disentangled representation, which is a challenging problem on its own. This is why we see  a performance drop going from 3 variables to 5 variables.
> > > > >
> > > > > - How a Feedback Transformer can solve it? It can store variable values in its memory activations. Because there are many vectors in the memory, it can store each variable in its own vector. This makes it easy to avoid the disentanglement problem that LSTMs suffered. Now, when there are inputs “x++”, a model can read the memory containing “x=4” at the 1st layer because memories feedback to all layers. Therefore, operation “x+1=5” can be performed at the 1st layer and write “x=5” to a newly created memory at the time step. Which means, even a very shallow Feedback Transformer can update variable values, and solve the task.
> > > > >
> > > > > **re question 2**: We haven’t done any large-scale RL task yet. Our focus was initially on investigating limitations in standard Transformers - after identifying a few in settings where Transformers are dominant architectures (mainly in the NLP domain), we worked to develop the Feedback architecture to remedy those limitations. We designed several tasks to challenge those limitations (including these RL gridworld tasks) to illustrate how they can easily be overcome with the straightforward modification of the Feedback architecture. We are very interested in trying larger-scale RL environments, but our focus is on proposing how to overcome natural limitations of the Transformer architecture --- we will investigate using Feedback to push state of the art in RL next!

---

> > > > > > ### Comment · AnonReviewer2 · 2020-11-21
> > > > > > **More response to the authors' response above**
> > > > > >
> > > > > > Regarding question 1, I guess I'm more interested in why an 8L Transformer is worse than a 4L Transformer. Because if the rationale behind the gap is truly as what the authors explained (i.e., more updates to the "belief state" are better, cf. Section 3.2), then I would expect 8-layer Transformers to do better than the 4-layer counterpart.
> > > > > >
> > > > > > Since being able to memorize the "starting value" of the variables is the important thing in the "algorithmic task", a common and yet extremely simple practice is simply to augment the memory of the Transformer. For example, you can just append a `[mem]` token to the input  & hidden sequences on the left (like in BERT, where you can augment the sequence information with extra `[CLS]` tokens as well), and then explicitly accumulate whatever information the model finds useful there. I guess a minor point is, in this particular task, there seems to be an absolute necessity/benefit to have cross-layer back-reference; and if you simply add things like skip connections or memory augmentations to a standard Transformer (which won't sacrifice its parallelism, unlike the feedback case), would it work well too?
> > > > > >
> > > > > > The phenomenon with LSTMs, as the authors identify, might be a different issue, but I also am not fully convinced whether the disentanglement is the problem here (I personally think LSTM is "intelligent" enough to separate three symbols). I wonder whether the authors think something like a temporal convolution would work better than LSTM in this case. For example, group convolutions seems a pretty natural thing to do to "segment" hidden units into multiple sections for such kind of update.

---

> > > > > > > ### Author Response · Authors · 2020-11-22
> > > > > > > **response to response :)**
> > > > > > >
> > > > > > > >Regarding question 1, I guess I'm more interested in why an 8L Transformer is worse than a 4L Transformer. Because if the rationale behind the gap is truly as what the authors explained (i.e., more updates to the "belief state" are better, cf. Section 3.2), then I would expect 8-layer Transformers to do better than the 4-layer counterpart.
> > > > > > >
> > > > > > > **Response:** The 8L Transformer is most likely worse because it has more capacity to overfit (after all, the training data only contains 10k codes).
> > > > > > >
> > > > > > > >Since being able to memorize the "starting value" of the variables is the important thing in the "algorithmic task", a common and yet extremely simple practice is simply to augment the memory of the Transformer. For example, you can just append a [mem] token to the input & hidden sequences on the left (like in BERT, where you can augment the sequence information with extra [CLS] tokens as well), and then explicitly accumulate whatever information the model finds useful there. I guess a minor point is, in this particular task, there seems to be an absolute necessity/benefit to have cross-layer back-reference; and if you simply add things like skip connections or memory augmentations to a standard Transformer (which won't sacrifice its parallelism, unlike the feedback case), would it work well too?
> > > > > > >
> > > > > > > **Response:** It is important to know the "current value" rather than the "initial value" in this task (because there are conditionals and print statements). When the "current value" is updated by "x++", a Transformer will need to read it from the k-th layer, update it, and store it at k+1-th layer. Therefore, the "current value" will move to a higher layer with each update, eventually reaching the top layer, then becoming forgotten. In general, this is true for any feedforward model with a fixed depth. It is not clear how the suggested modifications would work, but as long as a Transformer stays a feedforward model with a fixed depth, it will have this problem.
> > > > > > >
> > > > > > > >The phenomenon with LSTMs, as the authors identify, might be a different issue, but I also am not fully convinced whether the disentanglement is the problem here (I personally think LSTM is "intelligent" enough to separate three symbols). I wonder whether the authors think something like a temporal convolution would work better than LSTM in this case. For example, group convolutions seems a pretty natural thing to do to "segment" hidden units into multiple sections for such kind of update.
> > > > > > >
> > > > > > > **Response:** Yes, a LSTM worked well with 3 variables as our result shows. But it starts struggle when there are 5 variables. We haven’t looked into temporal convolutions in our work, but we would guess it will have the same problem as Transformers because it is a feedforward model.

---

> ### Author Response · Authors · 2020-11-19
> **any additional questions?**
>
> Let us know if there's anything that's unclear or any additional questions that we can answer before the discussion period ends.

---

> ### Comment · AnonReviewer2 · 2020-11-23
> **Thank you to the authors for the detailed response**
>
> I would like to thank the authors for the detailed response and patient discussion. After the various discussions with the authors, I found this paper still has certain flaws unresolved (e.g., I still share the same opinion with R4 on that the belief-state-related arguments are somewhat not convincing enough and in shortage of stronger empirical support; and the lack of any large-scale RL tasks, even though the authors say it's a "future work", makes the value of this architecture more incremental in a lot of sense). I also agree with R1 that both the architectural modifications and the empirical results feel a bit incremental. I **strongly** encourage the authors to apply this on a larger-scale RL application (you don't even need to try something too large, but something of even a reasonable size is lacking now). It would be a much better and more natural choice than the synthetic tasks here to evaluate some of the core issues of the Transformer that the authors identify.
>
> However, I do appreciate the authors' rebuttal efforts where some of my questions are answered in a detailed manner. For instance, although the training is still slower, I feel its efficiency in a reasonable range now, and there's generally a slight speedup in the generation scheme because the K and V are shared.
>
> I have (cautiously) updated my score to 6, though still noting the various concerns I have above.

---

### Official Review · AnonReviewer1 · 2020-10-28
**Interesting findings**

**Rating:** 7
**Confidence:** 5

**Review:**


The main topic of this paper is modification and enhancement of Transformers originally proposed in Vaswani’17.
As we all know, Transformers are now used as a core technology in a wide range of research communities such as natural language, vision, and speech.
Many researchers aim to improve such core technology since it might provide a high impact to the communities. Thus, tons of papers propose a wide variety of modifications for Transformers in recent years.
In this perspective, this paper can be categorized as one of such papers.
Therefore, the audience and influence of this paper could be significantly broader.


This paper focuses on the limitations of the Transformer architecture as an autoregressive model.
This paper points out two drawbacks.
One is the original Transformers do not handle the higher layer representations of the past states that have already been computed as a viewpoint of the autoregressive model.
The other is that the model depth bounds the number of transformations possible on the input.
This paper then proposes a method called “Feedback transformers“ that can effectively overcome such drawbacks by explicitly incorporating all the past state representations, including higher-layer representations, when using transformers as an auto-regressive sequential generator.

The top-level concept is rather straightforward and can be easily noticeable by many researchers in some sense, nothing innovative or unique.
From this perspective, it seems that this paper is incremental study rather than an innovative one.
However, the idea of injecting auto-regressive computation is the somewhat totally counter concept for the original Transformers since Transformers try to significantly reduce the computational cost on the specialized computational environment like GPUs by ignoring the auto-regressive nature.
Although the proposed method does not obey the original concept of Transformers, the findings from this paper's experiments are very impressive.
I think the findings in this paper can help many researchers as a new insight into the community.
In my feeling, this is basically an insightful paper.


The following are the questions/concerns of this paper.

1, The implicit explanation for the target situation
The discussion in this paper only focuses on the auto-regressive generation or sequentially predicting tokens one-by-one.
However, the Transformer architectures are also popular to be used in many other situations, such as masked language models like BERT.
Unfortunately, the current version does not explicitly distinguish how the Transformer architectures are used for.
Some readers might misunderstand that the discussion of this paper could include such a situation.
Even if not so much, the authors should clearly state the target of their claims and discussions at the very beginning of this paper.


2, Calculation cost
The calculation cost is one of the main discussion points in the proposed method.
However, there is no clear experimental results shown about this part.
This is a clear disadvantage of this paper.


3, Intuition of additional parameters w^l appeared in Eq. 1. (Ablation study)
The proposed method suggests using the weighted sum of all the hidden vectors in all the layers.
However, there is no reasonable explanation about intuition for this introduction.
We have many other possible choices.
This paper does not discuss such a possible variant at all.
To better understand the proposed method, the authors should provide a certain amount of ablation studies.

For example, what would happen if we used all the hidden vectors independently, not just weighted summing ups them.
Moreover, what would happen if just average them (without weighting factor), etc.





4,
I am not totally convinced that MT and LM's results are really significant improvements from the original Transformers or the comparative previous methods.
Regarding the WMT experiments, the one-point BLEU difference can be often observed by just changing random seeds in the identical method.
The authors should somehow provide additional evidence that the proposed method significantly differs from the baseline methods.


I am willing to change my score if I got reasonable answers for all the questions and concerns written in the above reviews.

---

> ### Author Response · Authors · 2020-11-17
> **review response**
>
> Thanks for your detailed review! We have responded to each point below and included several additional experimental results to answer your questions quantitatively. Let us know if you have additional questions or suggestions, thanks!
>
> **re: The implicit explanation for the target situation**
> In the Introduction, we state that we focus on these limitations only for Transformers as autoregressive models, which does not apply to BERT-style masked language model architectures. We added the sentence "These limitations and our proposed solution target sequential token prediction tasks, such as language modeling or other auto-regressive generative tasks." We also edited the abstract to change the first sentence to "Transformers have been successfully applied to sequential, auto-regressive tasks" and clarify this point.
>
> **re: Calculation cost**
> We include a table here about the training and validation speed of the Feedback Transformer compared to the standard Transformer for language modeling, translation, and reinforcement learning. For language modeling, we measure on Wikitext-103 (we compare a 8-layer Transformer against a 4-layer Feedback model of the same size. Both models have a fixed attention span of 512, and trained on 32GPUs. With the Feedback model, we’re able to fit 2x larger batches in GPU memory. The inference is done with 1GPU). For translation, we measure on WMT En-De with 6 layer encoder and 2 layer decoder (reporting training WPS on 8 GPU and inference WPS on 1 GPU). For RL, we report the training frame-per-second (FPS) on the maze navigation task (using 20 CPU cores and 1 GPU). We will add this table into the main paper.
>
> |Task |Model | Training WPS | Inference WPS `|
> | --- | --- | --- | --- |
> |LM (wiki103) |Transformer | 296K | 592 |
> |LM (wiki103) |Feedback |84.K | 2176 |
> |Translation |Transformer | 280K | 3190 |
> |Translation | Feedback |126K | 5410 |
> |RL Maze | Transformer | 22.3K | --- |
> |RL Maze |Feedback | 22.3K | --- |
>
> For Encoder-Decoder tasks, the Feedback Transformer is slower than the standard Transformer, but is faster at Inference as it uses less memory and can thus generate translations with larger batch sizes. For Language modeling, the Feedback Transformer is about 3x slower to train, but much faster at inference due to reduced memory cost (from sharing key-values) and reduced depth. For reinforcement learning tasks, the training must be online as well, so the Transformer and Feedback Transformer have the same speed.
>
> **re:  Intuition of additional parameters**
> We investigated different ways to compose the memory in Figure 5, which indicates that the Feedback Memory form is the strongest performing one. To answer your question about averaging, we added an additional experiment in Section 6.4 (see Figure 8). Representations from higher layers work better as memory, confirming our assumption of the importance of higher level representations in the Feedback Transformer. Simply averaging all layers together works reasonably well, but the weighted sum approach matches the best performance because it can adopt to select any of the layers.
>
> **re: Significant Improvements**
> For Wikitext-103, we report 18.3 PPL for the Feedback Transformer. If we use our codebase to run a standard Transformer, the performance is only 19.9 PPL, which is substantially worse. This baseline is new, and we have updated Table 3.
>
> We have conducted additional experiments to understand the variance on translation to answer your specific point about the importance of a 1-BLEU difference. On WMT En-De, we train three models with different seeds and see that the standard deviation in BLEU is around **0.15** for the baseline standard Transformer and around **0.12** for the Feedback Transformer. This standard deviation is fairly stable across models with varying decoder layers. We will display standard deviation in Figure 4 (left) in the main paper. We conclude based on this investigation that the improvement from Feedback Transformer --- much stronger performance with shallow models --- is statistically significant. For example, with 1 decoder layer, the Feedback Transformer has about 1 BLEU improvement. At full model size, the Feedback Transformer outperforms the standard Transformer, but marginally (0.2 BLEU improvement).

---

> ### Author Response · Authors · 2020-11-19
> **any additional questions?**
>
> Let us know if there's anything that's unclear or any additional questions that we can answer before the discussion period ends.

---

> > ### Comment · AnonReviewer1 · 2020-11-23
> > **Official Blind Review #1**
> >
> > Thank you for the detailed response to my concerns/questions.
> > Some of my concerns, such as computational cost and significance of the results, were answered by the response.
> >
> > Slow training is not a desired property, but a faster inference is much preferable.
> > I would like the authors to add an explanation of why the proposed method gets slow training than the baseline Transformers for clear discussion.
> >
> > I have also read other reviewers' reviews.
> > I understand that this paper has some unresolved flaws, and the novelty of the proposed method seems incremental, not innovative, as I wrote in my review.
> > However, I feel that this paper shows some new findings that can be useful for the community, and some flaws are not too much critical, in my opinion.
> > This paper should have a chance to be accepted to the conference, and thus, I increase my scores to 7.

---

### Author Response · Authors · 2020-11-17
**general response to all reviewers**

Thanks for the reviews. We are happy to see that the reviewers think that this is “good architectural improvement” (R2) that can achieve faster inference and stronger performance with smaller models (R3). We’re happy the reviewers think the “experiments are very impressive” (R1) which can “help many researchers as a new insight” (R1). We have updated our paper with additional experimental results to answer reviewer questions, and adjusted the text to add detail.

We would like to emphasize two general points:

**(1)** **Training Speed**- We show detailed results about training and inference speed, comparing similar model sizes. For language modeling, we measure on Wikitext-103 (we compare a 8-layer Transformer against a 4-layer Feedback model of the same size. Both models have a fixed attention span of 512, and trained on 32GPUs. With the Feedback model, we’re able to fit 2x larger batches in GPU memory. The inference is done with 1GPU). For translation, we measure on WMT En-De with 6 layer encoder and 2 layer decoder (reporting training WPS on 8 GPU and inference WPS on 1 GPU). For RL, we report the training frame-per-second (FPS) on the maze navigation task (using 20 CPU cores and 1 GPU). We will add this table into the main paper.


|Task                     |Model                | Training WPS	        | Inference WPS `|
| --- | --- | --- | --- |
|LM (wiki103)	  |Transformer	       | 296K	                           | 592 |
|LM (wiki103)      |Feedback 	              |84.K	                        | 2176 |
|Translation	     |Transformer	       | 280K	                      | 3190 |
|Translation       | Feedback 	          |126K	                       | 5410 |
|RL Maze	        | Transformer	    | 22.3K	                          | --- |
|RL Maze	          |Feedback 	        | 22.3K	                        | --- |

**Inference is actually faster** with the Feedback Transformer because our key-value project sharing substantially reduces the memory footprint of the model and reduces computation **plus** Feedback allows shallow models to perform very well, which allows us to increase batch size. In language modeling, for example, sharing key-value provides almost 3X inference speed improvement. The shallow model size provides the remaining 10% of speed improvement at inference time. Finally, note that for certain problems (such as in RL), the data must be processed strictly sequentially anyway and Feedback Transformer is not any slower.

We note that predominant architectures are often influenced by what runs quickly on existing hardware (see “The **Hardware Lottery**” https://arxiv.org/abs/2009.06489). Our work addresses two major limitations of the Transformer and proposes a simple solution to this that allows small, shallow models to have much stronger performance. Overall, we feel that there should be focus on “is this model interesting” and “does this model solve real limitations” --- if we only focus on initial implementations being inefficient, it limits the kinds of possible research directions.


**(2) Additional algorithmic task** - to address reviewer questions about belief state, we have added an algorithmic task that requires careful tracking to section 4.1.2. This task requires models to execute sequence of statements, which obviously cannot be processed in parallel because conditional statements cannot be executed without knowing the current variable value, which itself can depend on another conditional statement. **Transformers cannot solve** this task because every time a variable increment or decrement, its value can only be found one layer up in the model, and eventually will be lost. Making the model deeper does help a little, but their accuracy is far from LSTM, which is still outperformed by Feedback Transformer.


**New Baseline**: We added a Transformer baseline, using our codebase, to Wikitext-103 results in Table 3. The only difference between this baseline and our model is in how memory is composed, we can see more clearly the improvement brought by the Feedback memory, which is 1.6 PPL.

**Correction:** We made a small correction to the Fig. 5 where the “all” number is slightly increased.

---

### Decision · Program_Chairs · 2021-01-07
**Final Decision**

**Decision:**

Reject

**Comment:**

This paper focuses on the limitations of the transformer architecture as an autoregressive model. The paper is relatively easy to follow. Though most reviewers find the paper interesting, the idea is not very novel. The introduction of sequential-ness to Transformer is good, though it also slow things down especially as the sequence gets longer.

An extensive set of experiments are performed, though the results are not entirely convincing. The authors are encouraged to add more ablative experiments, efficiency analysis, and large-scale results.